# A Cu(II)–ATP complex efficiently catalyses enantioselective Diels–Alder reactions

Changhao Wang [1,3 ✉], Qianqian Qi[1,3], Wenying Li[1], Jingshuang Dang[1], Min Hao[1], Shuting Lv[1], Xingchen Dong[1], Youkun Gu[1], Peizhe Wu[1], Wenyue Zhang[1], Yashao Chen[1] & Jörg S. Hartig[2]

Natural biomolecules have been used extensively as chiral scaffolds that bind/surround metal complexes to achieve stereoselectivity in catalytic reactions. ATP is ubiquitously found in nature as an energy-storing molecule and can complex diverse metal cations. However, in biotic reactions ATP-metal complexes are thought to function mostly as co-substrates undergoing phosphoanhydride bond cleavage reactions rather than participating in catalytic mechanisms. Here, we report that a specific Cu(II)-ATP complex ($Cu^{2+}$·ATP) efficiently catalyses Diels-Alder reactions with high reactivity and enantioselectivity. We investigate the substrates and stereoselectivity of the reaction, characterise the catalyst by a range of physicochemical experiments and propose the reaction mechanism based on density functional theory (DFT) calculations. It is found that three key residues (N7, β-phosphate and γ-phosphate) in ATP are important for the efficient catalytic activity and stereocontrol via complexation of the Cu(II) ion. In addition to the potential technological uses, these findings could have general implications for the chemical selection of complex mixtures in prebiotic scenarios.

[1] Key Laboratory of Applied Surface and Colloid Chemistry, Ministry of Education, School of Chemistry and Chemical Engineering, Shaanxi Normal University, Xi'an, China. [2] Department of Chemistry and Konstanz Research School Chemical Biology (KoRS-CB), University of Konstanz, Konstanz, Germany. [3] These authors contributed equally: Changhao Wang, Qianqian Qi. ✉email: changhaowang@snnu.edu.cn

Artificial metalloenzymes (ArMs) are usually constructed by embedding metal cofactors into the chiral scaffolds of biological molecules that are used to expand the reaction types and unearth novel functions of the biomolecules. Over the past four decades, protein-based ArMs have been widely investigated to achieve a variety of valuable enantioselective transformations[1–6]. To precisely characterise the active centres and obtain insight into the reaction mechanisms of ArMs, simple scaffolds of peptides and amino acids have been employed to rationally design artificial metallo-peptides[7–13] and metallo-amino acids[14–21]. In recent years, nucleic acids have aroused much interest among chemists for constructing diverse nucleic acid-based ArMs for enantioselective catalysis. Natural double-stranded DNA (dsDNA) was first employed as a chiral scaffold in a supramolecular assembly with an achiral copper(II) complex, realising the chirality transfer from dsDNA to the products[22]. Since then, many synthetic dsDNAs have been designed to covalently anchor metallic moieties to produce DNA-based ArMs[23–28]. These artificial designs enable the fine-tuning of the microenvironment and provide a deeper understanding of the origin of chiral induction. Through their tuneable structures, G-quadruplexes containing 21- to 69-mer nucleotides have been employed to construct G-quadruplex DNA metalloenzymes, which have been successfully applied to several enantioselective transformations and demonstrated to depend largely on the conformation of the non-canonical G-quadruplex structure[29–36]. In addition, a short single-stranded 11 nt DNA within a G-triplex structure was shown to bind to copper(II) ions and modestly promote an enantioselective Diels–Alder (D–A) reaction[37].

In addition to DNA, a large number of ribozymes generated by in vitro selection exhibit catalytic activity with the assistance of metal ions[38]. Most importantly for the present work, a ribozyme was selected to catalyse a D–A reaction. The Diels–Alderase ribozyme shows a 20,000-fold rate enhancement and relies on the presence of divalent metal ions such as $Mg^{2+}$ or $Mn^{2+}$[39]. In a further work Jaeschke and co-workers developed a 49-mer Diels–Alderase ribozyme that could catalyse the D–A reaction of anthracene derivatives and maleimides with an enantiomeric excess (ee) of up to 95%[40]. Systematic experiments based on the anthracene derivative dienes and maleimide dienophiles suggest that the stereoselectivity of the reaction is mostly controlled by RNA–diene interactions and the hydrophobic side chain of the dienophile is responsible for RNA binding[41]. The crystal structure of the Diels–Alderase ribozyme shows that the $Mg^{2+}$ ion is a structural cofactor that stabilises the wedge shaped pocket and the stereoselectivity is governed by the shape of the catalytic pocket[42].

In addition to in vitro-selected ribozymes such as the Diels–Alderase, synthetic double-stranded RNAs (dsRNAs) were shown to interact with copper(II) complexes to form RNA-based ArMs, giving rise to high reactivity yet modest enantioselectivity in a Friedel–Crafts reaction[43]. Another RNA-based ArMs containing either dsRNA or hairpin RNA exhibited very low enantioselective induction in a D–A reaction compared with the corresponding DNA-based ArMs[44]. The current nucleic acid-based ArMs always contain several tens to hundreds of nucleotides to achieve chiral scaffolding. In most cases, the precise location of the catalytic metal species is unclear, and high resolution structures of these ArMs are lacking. Recently, our group reported a cyclic dinucleotide (c-di-AMP)-based artificial metalloribozyme that catalyses a Friedel–Crafts reaction with high enantioselectivity[45]. Furthermore, a phosphine-modified deoxyuridine coordinating a palladium species enables an enantioselective allylic amination[46], suggesting that the chirality of the sugar could be transferred to the product. Therefore, the design of nucleic acid-based ArMs with only a few nucleotides as the scaffold appears to be a promising approach for obtaining

minimal systems that might be better suited for gaining accurate structural information and providing deeper insights into the reaction mechanisms. ATP is a well-known energy-storing molecule that participates in many processes in living organisms. In most enzyme-catalysed reactions, ATP acts as a co-substrate undergoing phosphoanhydride bond cleavage reactions. However, ATP has been demonstrated to specifically bind metal ions with high affinity and could therefore function as a chiral scaffold participating in enantioselective catalysis mediated by the complexed metal ions.

In this work, it is found that ATP interacts with $Cu^{2+}$ ions to form a Cu(II)-ATP complex ($Cu^{2+}\cdot ATP$) that efficiently catalyses enantioselective D–A reactions (Fig. 1). From the ATP analogues experiments and spectroscopic characterisations, the purine moiety, β- and γ-phosphates in ATP are revealed as vital residues to coordinate $Cu^{2+}$ ion for exerting enantioselective catalytic activity. The theoretical calculations further support a fine coordination structure of $Cu^{2+}\cdot ATP$, in which the $Cu^{2+}$ ion binds to N7 atom from adenine and two oxygen atoms from β- and γ-phosphates, together with a hydrogen bond between 6-amino and γ-phosphate oxygen. This work demonstrates that a single nucleoside polyphosphate is sufficient for chiral induction in enantioselective reactions and the catalytic function of ATP-metal complexes might implicate the chemical selection in primordial chemistry.

## Results

**Enantioselective Diels–Alder reactions catalysed by $Cu^{2+}\cdot ATP$.** A benchmark D–A reaction of azachalcone (**1a**) and cyclopentadiene (**2**) was employed to test the catalytic performance of ATP–metal ion complexes. The initial attempt using either ATP or a $Mg^{2+}\cdot ATP$ complex resulted in a very low conversion and ee value (Table 1, entries 1 and 2). To our delight, when copper(II) nitrate was added to ATP as a metal cofactor, the corresponding D–A reaction gave a conversion of 90%, an ee of **3a** (exo) of 74% and an ee of **3a** (endo) of 65% (Table 1, entry 4). These results indicate that ATP and $Cu^{2+}$ ion specifically interact to form a Cu(II)–ATP complex of $Cu^{2+}\cdot ATP$. Using other divalent metal ions such as $Zn^{2+}$, $Co^{2+}$ and $Ni^{2+}$ as the metal cofactors resulted in low conversions and ee values (Table 1, entries 5-7). The introduction of extra achiral ligands significantly inhibited the reaction catalysed by $Cu^{2+}\cdot ATP$ and **3a** was obtained with ee values of 29–46% (Table 1, entries 8–10). The distinct circular dichroism (CD) spectra of ATP in the presence of different copper(II) complexes (Supplementary Fig. S1) suggest that the presence of additional ligands might block the interaction between **1a** and the $Cu^{2+}$ ion, leading to poor reactivity and low enantioselectivity. Different copper(II) salts and molar ratios of ATP/$Cu^{2+}$ were screened (Supplementary Tables S1 and S2) and an ATP/Cu(OTf)$_2$ ratio of 5:1 resulted in optimal catalysis with 98% conversion and 72% ee in favour of the endo isomer (Table 1, entry 11). The addition of 10 mM $Mg^{2+}$ ions resulted in a significantly reduced ee value compared with that obtained from the $Cu^{2+}\cdot ATP$ catalysed D–A reaction (Table 1, entry 12), whereas $Na^+$ and $K^+$ ions caused no significant difference (Supplementary Table 3). Because $Mg^{2+}$ ions are known to bind to ATP[47–50], this result indicates that the presence of $Mg^{2+}$ might compete with the binding of $Cu^{2+}$ to ATP. Concerning the reaction temperature, the ee value decreased at an elevated temperature (Table 1, entry 11 vs. entry 13).

To investigate the substrate specificity of the $Cu^{2+}\cdot ATP$ catalyst, different azachalcones (**1b–h**) were investigated and modest to good stereoselectivities were obtained (Table 1, entries 14–20). Compared with **1a**, azachalcone **1b** or **1c**, which bears an electron-donating group (4-Me or 4-OMe) on the phenyl moiety,

**Fig. 1 Schematic representation of enantioselective Diels–Alder reactions catalysed by a Cu(II)-ATP complex.** The Cu(II)–ATP complex (Cu$^{2+}$·ATP) is composed of copper(II) trifluoromethanesulfonate (Cu(OTf)$_2$) and ATP. The coordination structure of Cu$^{2+}$·ATP is supported by spectroscopic characterisations and theoretical calculations, in which Cu$^{2+}$ ion binds to one nitrogen atom (N7) from adenine, one oxygen atom from β-phosphate, one oxygen atom from γ-phosphate and one trifluoromethanesulfonate anion (OTf$^-$) and simultaneously an intramolecular hydrogen bond is formed between a hydrogen atom from 6-amino and an oxygen atom from γ-phosphate. Typical Diels–Alder reactions of azachalcones (**1a–h**) and cyclopentadiene (**2**) are selected in this study, and the detailed reaction procedure is described in "Methods" section.

**Table 1 Enantioselective Diels–Alter reactions catalysed by metallo–ATP complexes[a].**

| Entry | Azachalcone | Metal cofactor | Conversion (%) | *Endo/exo* | ee (%, *exo*) | ee (%, *endo*) |
|---|---|---|---|---|---|---|
| 1 | **1a** | None | 3 | 85:15 | 29 | 16 |
| 2 | **1a** | MgCl$_2$ | 3 | 84:16 | 22 | 5 |
| 3[b] | **1a** | Cu(NO$_3$)$_2$ | 70 | 92:8 | 0 | 0 |
| 4 | **1a** | Cu(NO$_3$)$_2$ | 90 | 91:9 | 74 | 65 |
| 5 | **1a** | Zn(NO$_3$)$_2$ | 3 | 84:16 | 28 | 13 |
| 6 | **1a** | Co(NO$_3$)$_2$ | 11 | 85:15 | 31 | 13 |
| 7 | **1a** | Ni(NO$_3$)$_2$ | 23 | 87:13 | 44 | 35 |
| 8 | **1a** | Cu(bpy)(NO$_3$)$_2$ | 9 | 89:11 | 42 | 33 |
| 9 | **1a** | Cu(dmbpy)(NO$_3$)$_2$ | 5 | 87:13 | 34 | 29 |
| 10 | **1a** | Cu(phen)(NO$_3$)$_2$ | 8 | 90:10 | 48 | 46 |
| 11 | **1a** | Cu(OTf)$_2$ | 98 | 91:9 | 79 | 72 |
| 12[c] | **1a** | Cu(OTf)$_2$/MgCl$_2$ | 90 | 91:9 | 66 | 53 |
| 13[d] | **1a** | Cu(OTf)$_2$ | 71 | 89:11 | 48 | 44 |
| 14 | **1b** | Cu(OTf)$_2$ | 90 | 92:8 | 88 | 80 |
| 15 | **1c** | Cu(OTf)$_2$ | 85 | 92:8 | 80 | 77 |
| 16 | **1d** | Cu(OTf)$_2$ | 72 | 96:4 | 62 | 50 |
| 17 | **1e** | Cu(OTf)$_2$ | 80 | 91:9 | 93 | 39 |
| 18 | **1f** | Cu(OTf)$_2$ | 85 | 84:16 | 76 | 84 |
| 19 | **1g** | Cu(OTf)$_2$ | 91 | 91:9 | 56 | 44 |
| 20 | **1h** | Cu(OTf)$_2$ | 80 | 92:8 | 80 | 67 |

*bpy* 2,2′-bipyridine, *dmbpy* 4,4′-dimethyl-2,2′-bipyridine, *phen* 1,10-phenanthroline.
[a]Reaction conditions: **1** (1 mM), **2** (200 mM), ATP (250 μM), metal cofactor (50 μM), MES buffer (20 mM, pH 5.5), 4 °C, 24 h for **1a** and 72 h for **1b–h**. The conversion of **1a** was calculated by HPLC and the conversions of **1b–h** were determined from the crude products by $^1$H NMR. The diastereoselectivity (*endo/exo*) and enantioselectivity were determined from the crude products by chiral HPLC. All data were the averages of at least two individual experiments (reproducibility: ±5% conversion, ±3% *endo/exo* and ±3% ee).
[b]Without ATP.
[c]10 mM MgCl$_2$ was added.
[d]Reaction at 37 °C.

exhibited an enhanced ee in the corresponding reaction (Table 1, entry 11 vs. entries 14 and 15). Changing the methoxy substitution from the 4′-position to the 2′-position on the phenyl moiety of azachalcone (**1d**) caused a significant decrease in the reactivity and enantioselectivity (Table 1, entry 15 vs. entry 16). For the corresponding D–A reactions using azachalcones with electron-withdrawing moieties (**1e** and **1f**), **1e** bearing R = (4-Cl)C$_6$H$_4$ reacted with **2** to give a 93% ee in favour of the *exo* isomer of **3e** (Table 1, entry 17), whereas **1f** with R = (4-NO$_2$) C$_6$H$_4$ yielded an 84% ee in favour of the *endo* isomer of **3f** (Table 1, entry 18). In addition, good reactivity and modest enantioselectivity were achieved using azachalcones **1g** and **1h** with heterocyclic substitutions (Table 1, entries 19 and 20). These results suggest that the steric and electronic effects of the substituents of azachalcones greatly affect the catalytic performance of Cu$^{2+}$·ATP. In addition, the reaction of substrate **1a**

(105 mg, 0.5 mmol) was conducted at a large scale to test its potential practicability. At a Cu$^{2+}$·ATP loading of 5 mol%, the product **3a** was obtained with an isolated yield of 80% and an ee value of 65% in favour of the *endo* isomer. Overall, it was demonstrated that ATP and the Cu$^{2+}$ ion form a potent and practical entity of Cu$^{2+}$·ATP for enantioselective D–A reactions.

**Kinetics of Cu$^{2+}$·ATP catalysis.** To clarify the catalytic roles of ATP and Cu$^{2+}$, the apparent second-order rate constants ($k_{app}$) of ATP, Cu$^{2+}$ and Cu$^{2+}$·ATP were determined by monitoring the UV-Vis absorption of **1a** during the corresponding D–A reactions. Compared with the D–A reaction without a catalyst ($k_{app,uncat}$), that with ATP had a comparable $k_{app,ATP}$ (Table 2, entry 1 vs. entry 2), suggesting that ATP is not the catalytic species, as indicated in Table 1. Cu$^{2+}$ ions are efficient Lewis acid catalysts

for the D–A reaction and led to an approximately sevenfold rate acceleration (Table 2, entry 3). When ATP and $Cu^{2+}$ ions were present as the $Cu^{2+}\cdot$ATP complex, a 13-fold rate enhancement relative to that of the uncatalysed reaction was observed (Table 2, entry 4). The kinetic parameters suggest that ATP and $Cu^{2+}$ ions indeed interact to assemble into a Cu(II)–ATP complex, where ATP serves as the chiral scaffold and the $Cu^{2+}$ ions serve as the catalytically active species.

**Catalytic performance of ATP analogues**. To probe the binding sites of $Cu^{2+}$ ion to ATP essential for catalysis, ATP was replaced with different ATP analogues in the $Cu^{2+}\cdot$ATP catalysed benchmark D–A reaction. Compared with ATP with three phosphates as the scaffold, ADP and $Cu^{2+}$ ions catalysed the D–A reaction with a similar conversion but a sharply decreased ee value of 19% (Fig. 2). The further removal of phosphate to either AMP or adenosine resulted in a reduced conversion and racemic product **3a** (Fig. 2). To investigate whether ATP is decomposed to ADP or AMP in the $Cu^{2+}\cdot$ATP catalysed D–A reaction, the reaction medium was analysed by high-performance liquid chromato-

graphy (HPLC) and ATP remained nearly unchanged during the reaction at pH 5.5 (Supplementary Fig. S6). These results suggest that the triphosphates are indispensable for $Cu^{2+}\cdot$ATP and the β- and γ-phosphates of ATP are probably the binding sites for the $Cu^{2+}$ ions as hypothesised previously[51–54]. Furthermore, the ribose of ATP was changed to deoxyribose by using dATP. A significant reduction in both the reactivity and enantioselectivity was observed with $Cu^{2+}\cdot$dATP (Fig. 2). This result indicates that 2′-OH affects the catalytic performance of $Cu^{2+}\cdot$ATP and that the altered sugar conformation is less able to facilitate the reaction (Supplementary Fig. S7). In addition, several nucleobase analogues of ATP were tested. Compared with $Cu^{2+}\cdot$ATP, $Cu^{2+}\cdot$GTP provided **3a** with a comparable 63% ee, probably owing to the presence of all the residues in ATP critical for efficient catalysis as identified above (Fig. 2). However, $Cu^{2+}\cdot$UTP and $Cu^{2+}\cdot$CTP generated **3a** with significantly decreased ee values (Fig. 2), which might be attributed to changes in the chiral complex structure compared with that of $Cu^{2+}\cdot$ATP (Supplementary Fig. S8). These results indicate that the purine moiety in ATP analogues is also important for achieving enantioselective catalysis with $Cu^{2+}$ complexes.

**Physicochemical characterisations of the Cu(II)–ATP complex**. To investigate the interaction between ATP and $Cu^{2+}$ ions, several characterisation techniques were employed. The addition of $Cu^{2+}$ caused slight changes to the CD spectrum of ATP (Supplementary Fig. S9), indicating that $Cu^{2+}$ ions hardly change the conformation of ATP. To assess the binding affinity of the Cu (II)–ATP complex, the apparent binding constant ($k_b$) was determined by a UV titration experiment. The $k_b$ value for ATP and $Cu^{2+}$ ions was estimated to be $(3.2 \pm 0.1) \times 10^5\,M^{-1}$ based on the curve fitting (Fig. 3a), demonstrating that $Cu^{2+}$ ions have a high affinity for ATP. Because $Cu^{2+}$ ions are paramagnetic

**Table 2 Kinetic study of $Cu^{2+}\cdot$ATP.**

| Entry[a] | Catalyst | $k_{app}$ ($M^{-1}s^{-1}$) | $k_{rel}$ |
|---|---|---|---|
| 1 | None | $(1.6 \pm 0.1) \times 10^{-3}$ | 1.0 |
| 2 | ATP | $(1.3 \pm 0.3) \times 10^{-3}$ | 0.8 |
| 3 | $Cu(OTf)_2$ | $(1.1 \pm 0.1) \times 10^{-2}$ | 6.9 |
| 4 | $Cu^{2+}\cdot$ATP | $(2.0 \pm 0.1) \times 10^{-2}$ | 12.5 |

[a]Reaction conditions: **1a** (20–50 μM), **2** (5 mM), ATP (250 μM), $Cu(OTf)_2$ (50 μM), MES buffer (2 mL, 20 mM, pH 5.5), 4 °C. All kinetic measurements were performed in triplicate and standard deviations were calculated. The rate acceleration $k_{rel}$ was calculated as the ratio of $k_{app,cat}/k_{app,uncat}$, where $k_{app,uncat}$ and $k_{app,cat}$ are the $k_{app}$ values in the absence and presence of the catalyst, respectively.

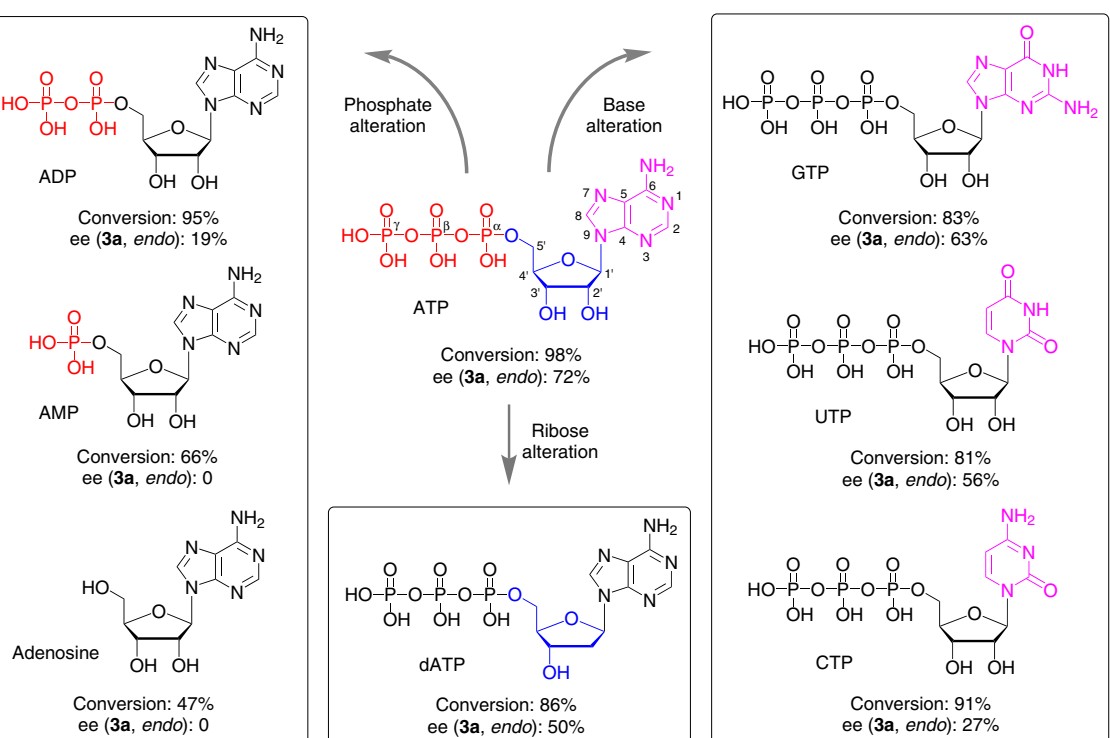

**Fig. 2 The effect of ATP analogues for the $Cu^{2+}\cdot$ATP-catalysed Diels–Alder reaction.** Reaction conditions: **1a** (1 mM), **2** (200 mM), ATP analogue (250 μM), $Cu(OTf)_2$ (50 μM), MES buffer (20 mM, pH 5.5), 4 °C, 24 h. All data are the averages of at least two individual experiments (reproducibility: ±5% conversion, ±3% ee).

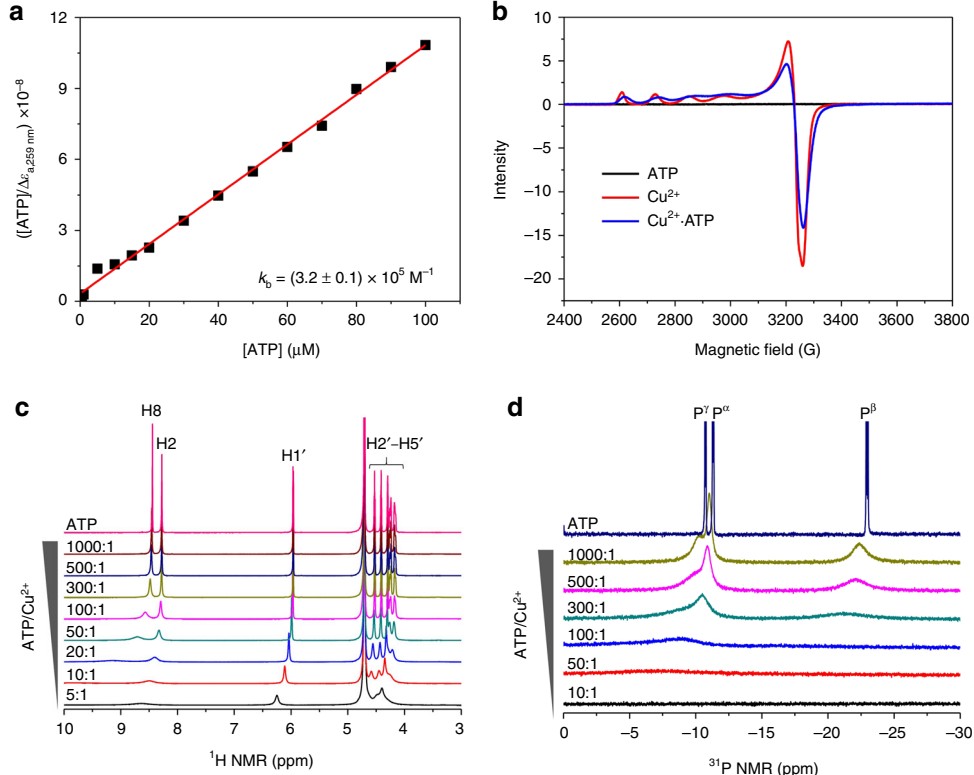

**Fig. 3 The interaction between ATP and Cu$^{2+}$ ions. a** The apparent binding constant ($k_b$) of ATP and Cu$^{2+}$ ions derived from the plot of [ATP]/$\Delta\varepsilon_{a,259\,nm}$ vs. [ATP] at various concentrations (5–100 μM) for Cu(OTf)$_2$ (5 μM). **b** EPR spectra of ATP, Cu$^{2+}$ and Cu$^{2+}\cdot$ATP. Experimental conditions: ATP (50 mM), Cu(OTf)$_2$ (10 mM), glycerol (20 v/v%), MES (20 mM, pH 5.5), 100 K. **c** $^1$H NMR and **d** $^{31}$P NMR spectroscopic titrations of ATP (120 mM) by varying the concentration of CuCl$_2$ (0.12–24 mM) in D$_2$O.

species, electron paramagnetic resonance (EPR) was employed to verify the interaction between ATP and Cu$^{2+}$ ions. The EPR spectrum of only Cu$^{2+}$ showed the typical fine structures of paramagnetic Cu$^{2+}$ ions (Fig. 3b). When ATP was added to Cu$^{2+}$ ions to form Cu$^{2+}\cdot$ATP, the fine structures of the Cu$^{2+}$ ions changed significantly and shifted (Fig. 3b), indicating that the Cu$^{2+}$ ions and ATP indeed interact. Furthermore, nuclear magnetic resonance (NMR) was used to detect possible binding sites of Cu$^{2+}$ ions in ATP. Owing to the paramagnetic property of Cu$^{2+}$ ions, the addition of Cu$^{2+}$ ions to ATP causes NMR signal broadening and shifting. When the ATP/Cu$^{2+}$ ratio was 300:1, the signal of H8 first broadened and shifted downfield relative to that in the $^1$H NMR of ATP, and further increasing the amount of Cu$^{2+}$ ions caused the H8 signal to broaden to the baseline and shift downfield (Fig. 3c). The H2 signals exhibited a similar but delayed tendency upon the addition of Cu$^{2+}$ ions to ATP (Fig. 3c). These results are in agreement with the reported interaction between ATP and Yb$^{3+}$ ions[55], indicating that Cu$^{2+}$ ions possibly bind to N7 in ATP. The chemical shifts of H1′–H5′ on the ribose ring remained nearly unchanged, even at an ATP/Cu$^{2+}$ ratio of 50:1 (Fig. 3c), indicating that these protons might be farther away from the paramagnetic centre of Cu$^{2+}$ ion. Compared with $^1$H NMR titration, the $^{31}$P NMR titration of ATP resulted in significant changes upon the addition of Cu$^{2+}$ ions. When 0.1% Cu$^{2+}$ ions were added to ATP, the P$^\beta$ and P$^\gamma$ signals immediately broadened and shifted downfield, and further increasing the ATP/Cu$^{2+}$ ratio to 100:1 led to the disappearance of the P$^\beta$ and P$^\gamma$ signals and the broadening and downfield shifting of the P$^\alpha$ signal (Fig. 3d). These results indicate that P$^\beta$ and P$^\gamma$ are sensitive to the presence of Cu$^{2+}$ ions, which probably interact with P$^\beta$-O and P$^\gamma$-O in accordance with the calculated

structure of Mg$^{2+}\cdot$ATP, see below. In short, the characterisation data together with the results for the above ATP analogues demonstrate that ATP and Cu$^{2+}$ ions interact via the β- and γ-phosphates as well as the purine N7. These binding sites for Cu$^{2+}$ ions have also been suggested in the literatures[54,56].

**Cu$^{2+}\cdot$ATP complex calculation and proposed reaction mechanism.** To further substantiate the hypothesised fine structure of the catalytically active Cu$^{2+}\cdot$ATP complex and explore a potential reaction mechanism, density functional theory (DFT) calculations were performed. Based on several proposed Cu$^{2+}\cdot$ATP models in the literature[56], a stable structure of Cu$^{2+}\cdot$ATP was obtained by a gas-phase calculation in which the Cu$^{2+}$ ion is bound to the N7 atom of adenine and the β- and γ-phosphate oxygen atoms and accompanying a hydrogen bond P$^\gamma$-O⋯H-N6 (Fig. 4a). The relative electronic energy ($\Delta E$) of the optimised Cu$^{2+}\cdot$ATP structure was 0.3 kcal mol$^{-1}$ lower than that of a previously described model obtained by a molecular orbital method[57] and 8.7 kcal mol$^{-1}$ lower than that of the Cu$^{2+}\cdot$ATP model without a hydrogen bond (Supplementary Fig. S22). With this Cu$^{2+}\cdot$ATP model in hand, the relative electronic energies of the precursor complexes of **1a**-Cu$^{2+}\cdot$ATP and **2** were calculated. Using the major product **3a** (*endo*) as an example, the $\Delta E$ value of the precursor of **1a**-Cu$^{2+}\cdot$ATP and **2** that yielded **3a** (*endo*) via the attack of the *Si* face was 9.1 kcal mol$^{-1}$ lower than that of the precursor for the *Re* face attack (Fig. 4b). The *Si* face attack was favoured owing to the hydrogen bonding between cyclopentadiene **2** and the phosphate oxygen atoms (Fig. 4b). This result indicates that the Cu$^{2+}\cdot$ATP-catalysed D–A reaction gives **3a** (*Si-endo*) with the preferred configuration of 1*R*, 2*S*, 3*S*, 4*S* (Supplementary Fig. S24), which is accordance with the experimental results

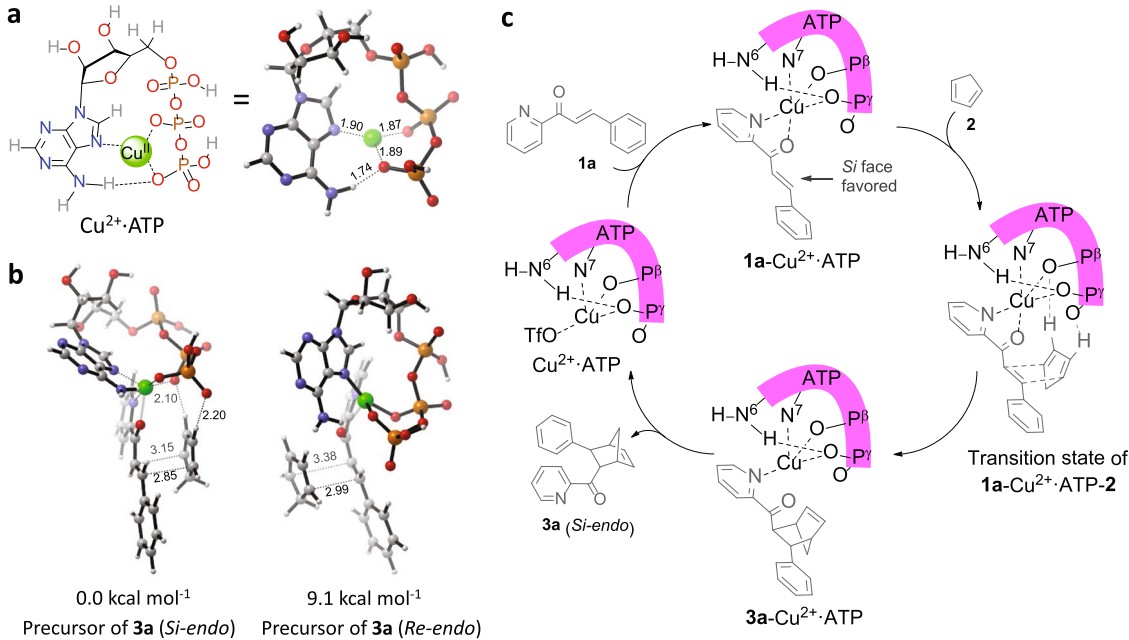

**Fig. 4 DFT calculations and a plausible reaction mechanism. a** Coordination structures of $Cu^{2+}$·ATP. For clarity, one trifluoromethanesulfonate anion coordinating to the copper(II) ion is omitted in $Cu^{2+}$·ATP. **b** Relative electronic energies ($\Delta E$) of the precursors of **3a** (*endo*) with **2** attacking the intermediate **1a**-$Cu^{2+}$·ATP from the *Si* and *Re* faces. **c** Plausible mechanism of the $Cu^{2+}$·ATP catalysed Diels–Alder reaction of **1a** with **2** yielding **3a** (*endo*) showing the favoured *Si* face attack of the intermediate of **1a** and $Cu^{2+}$·ATP.

(Supplementary Figs. S20, S21). The $\Delta E$ values of the precursors and products of **3a** (*exo*) were also calculated. The precursor of **3a** (*Si-exo*) was more stable than that of **3a** (*Re-exo*), further suggesting that cyclopentadiene **2** favoured the attack of **1a**-$Cu^{2+}$·ATP from the *Si* face (Supplementary Fig. S23). However, the $\Delta E$ value of **3a** (*Re-exo*) was 3.1 kcal mol$^{-1}$ lower than that of **3a** (*Si-exo*), in accordance with the experimental results (Supplementary Figs. S21, S24). Based on the experimental and theoretical results, a plausible reaction mechanism was proposed (Fig. 4c). The addition of **1a** to the stable $Cu^{2+}$·ATP catalyst gave rise to the intermediate **1a**-$Cu^{2+}$·ATP, which was in a pentacoordination state with newly formed Cu···N(**1a**) and Cu···O(**1a**) interactions. Because of the hydrogen bonding between ATP and **2**, **2** preferentially attacked the *Si* face of **1a** in the intermediate of **1a**-$Cu^{2+}$·ATP, leading to a relatively stable transition state consisting of $Cu^{2+}$·ATP, **1a** and **2**. The transition state of **1a**-$Cu^{2+}$·ATP-**2** automatically converted to the intermediate of **3a**-$Cu^{2+}$·ATP, which was accompanied by the breaking of the Cu···O(**1a**) bond, and the major D–A product **3a** (*Si-endo*) was obtained after release from $Cu^{2+}$·ATP.

## Discussion

In summary, we discovered that an enantioselective catalyst composed of the single nucleoside triphosphate ATP in complex with $Cu^{2+}$ ions is able to catalyse a D–A reaction with significant rate acceleration and high enantioselectivity. The purine structure and the phosphates at the β- and γ- positions are vital factors contributing to the enantioselective activity of the $Cu^{2+}$·ATP catalyst. Based on control experiments, physicochemical characterisations, and DFT calculations, a fine coordination structure of $Cu^{2+}$·ATP in which the $Cu^{2+}$ ion binds to the N7 atom, β-phosphate oxygen atom, and γ-phosphate oxygen atom and an intramolecular hydrogen bond between the 6-amino and γ-phosphate oxygen moieties was proposed. Of the reported metallo-biohybrid catalysts for enantioselective D–A reactions (Supplementary Table S6), the $Cu^{2+}$·ATP catalyst reported here is competitive, especially taking

into account the much simpler chiral scaffold compared to DNA, RNA and proteins. Importantly, compared to ADP and AMP, ATP proved to be a superior ligand for $Cu^{2+}$ binding and formation of a potent Cu(II)–ATP complex. The proposed fine coordination structure of the complex is able to explain the origin of chiral induction and should facilitate the rational design of further simple but efficient nucleotide-based catalysts.

In addition to its potential use in synthesis approaches, this work suggests that nucleotides could have played a role in the chemical selection of complex mixtures in prebiotic reactions in early evolutionary scenarios. The observation that several of the currently ubiquitous cofactors involved in redox and C–C bond formation reactions (e.g. nicotinamide-, flavine-, pantotheine- and cobalamine-based cofactors) contain adenine nucleosides and nucleotides could indicate a more pronounced role of ATP as ligand in prebiotic reactions. Although the D–A reaction described in this work is not proposed to be highly important in early chemical evolution, aldol reactions are thought to have played a crucial role in the establishment of early metabolic pathways[58], even providing early access to nucleotides from very simple starting materials[59]. However, how mirror symmetry breaking in aldol and other reactions occurred is still an open question[60,61]. Since aldol reactions are also catalysed by divalent metal ions[62–65], it is possible that nucleotide polyphosphates have played a role in the chemical selection of important metabolites in the abiotic stages of the emergence of life. Experiments investigating the potential of nucleotides to initiate chiral induction in aldol reactions are currently underway in our lab.

## Methods

**Typical procedure for $Cu^{2+}$·ATP catalysed D–A reactions.** A stock solution of ATP in water (final conc. 250 μM) and a freshly prepared aqueous solution of Cu (OTf)$_2$ (final conc. 50 μM) were added to an MES buffer solution (20 mM, pH 5.5) in a 10 mL vial to a total volume of 1000 μL. After stirring for 30 min at 4 °C, a thoroughly mixed solution of azachalcone **1a** (10 μL of a 0.1 M stock solution in CH$_3$CN, 1 μmol) and freshly distilled cyclopentadiene **2** (16 μL, 200 μmol) were immediately added. After the mixture was stirred for 24 h at 4 °C, the aqueous media were extracted by diethyl ether (3 × 2 mL) and flushed through a short gel column (a 5 cm length of glass dropper was filled with the silica gel to a height of

ca. 2 cm with some cotton at the bottom). The combined organic layers were removed under reduced pressure and the residue was directly analysed by chiral HPLC using a Daicel Chiralpak ODH column ($250 \times 4.6$ mm) with hexane and isopropanol as the eluents. The conversion of **1a** was calculated using the following Eq. (1):

$$\text{Conversion of } \mathbf{1a}(\%) = A_{\mathbf{3a}}/(A_{\mathbf{3a}} + A_{\mathbf{1a}}/f), \tag{1}$$

where $A_{\mathbf{1a}}$ and $A_{\mathbf{3a}}$ are the HPLC areas of **1a** and **3a**, respectively. The relative correction factor $f$ was 0.595.

**Kinetic assays for Cu$^{2+}$·ATP**. All kinetic measurements were performed by monitoring the disappearance of the absorption of **1a** at 326 nm, followed by the reference[66], using UV–Vis spectroscopy at 4 °C. ATP (final conc. 250 μM) in an MES buffer (20 mM, pH 5.5) was added to a 2 mL quartz cuvette containing a small magnet and stirred for 10 min, and then an aqueous solution of Cu(OTf)$_2$ (final conc. 50 μM) was added. After stirring for another 20 min, a fresh solution of **1a** (4, 6 or 10 μL of 0.1 M stock solution in CH$_3$CN) was added. Followed by an immediate addition of **2** (final conc. 5 mM), the measurement was started and the cuvette was sealed tightly. The initial rate ($V_{\text{init}}$) was determined from the slope of the line fitted to the decrease in the absorption of **1a** versus time, and the following Eq. (2) was used to calculate $V_{\text{init}}$:

$$V_{\text{init}} = d[A_{\mathbf{1a}}]/dt \cdot (d \cdot (\varepsilon_{\mathbf{1a}} - \varepsilon_{\mathbf{3a}}))^{-1}, \tag{2}$$

where $d[A_{\mathbf{1a}}]/dt$ is the slope of the absorption of **1a** vs. time during the initial 15% of the reaction, and $d$ is the path length of the cuvette. The parameters $\varepsilon_{\mathbf{1a}}$ and $\varepsilon_{\mathbf{3a}}$ are the molar extinction coefficients of **1a** and **3a**, respectively (Supplementary Figs. S13, S14). All parameters were measured at least three times.

The apparent second-order rate constant ($k_{\text{app}}$) was determined according to the procedure described in the literature[66]. The following Eq. (3) was used to calculate $k_{\text{app}}$:

$$k_{\text{app}} = d[A_{\mathbf{1a}}]/dt \cdot \left(d \cdot (\varepsilon_{\mathbf{1a}} - \varepsilon_{\mathbf{3a}}) \cdot [\mathbf{1a}]_0 \cdot [\mathbf{2}]_0\right)^{-1} = V_{\text{init}}/([\mathbf{1a}]_0 \cdot [\mathbf{2}]_0). \tag{3}$$

where $[\mathbf{1a}]_0$ and $[\mathbf{2}]_0$ are the initial concentrations of **1a** and **2**, respectively.

**DFT calculations**. All calculations of the reactions were performed in the gas-phase with Gaussian 16[67]. The molecular geometries of the precursors, transition states, and products were optimised at the B3LYP-D3/LANL2DZ ~3–21G level; the 3–21G basis set was used for C, H, O, N, and P, whereas LANL2DZ was used for Cu. Then, single-point energy corrections were obtained using M06-2X-D3/LANL2DZ ~6–311 G(d,p); the 6–311G(d,p) basis set was used for C, H, O, N and P, whereas LANL2DZ was used for Cu. The kinetic barriers of the non-catalytic reaction and the catalytic reaction with the most stable precursor were evaluated by calculating the single-point energies at the M06-2X-D3/LANL2DZ ~6–311G(d,p) level. All the structures were verified to be local minima by frequency calculations, whereas all the transition state species had only one imaginary frequency.

## Data availability
The data supporting the findings of this work are available within the article and its Supplementary Information files. All other relevant data of this study are available from the corresponding author upon reasonable request. Source data are provided with this paper.

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

## Acknowledgements
We are grateful for the financial supports of the National Natural Science Foundation of China (Nos. 21703132, 21773149, 21273142), the Natural Science Foundation of Shaanxi Province of China (2019JQ161) and the Fundamental Research Funds for the Central Universities (GK201802001). We thank Dr. Xinai Guo for technical assistance for the EPR characterisation.

## Author contributions
C.W. and Q.Q. contributed equally to this work. C.W. conceived and directed the project, analysed the data, wrote and revised the manuscript. Q.Q. performed the experiments, analysed the data and prepared the Supplementary Information. W.L. and J.D. carried out the DFT calculations. M.H., S.L., X.D., Y.G., P.W. and W.Z. conducted the spectroscopic characterisations. Y.C. directed the project and revised the manuscript. J.S.H revised and commented the manuscript.

## Competing interests
The authors declare no competing interests.
