## [Peer Review File · Nature Communications]

Reviewer #1 (Remarks to the Author):

The manuscript "A Cu(II)-ATP complex efficiently catalyzes enantioselective Diels-Alder reactions" by Wang et al. describes the preparation and characterization of ATP metal complexes. Catalysis of the Diels-Alder cycloaddition of aza-chlacones and cyclopentadiene by the ATP-Copper(II) complex was efficient and stereoselective. The authors then characterized the ATP-Copper(II) complexation using various techniques and proposed a reaction mechanism that was sustained by DFT calculations.

After carefully examining this manuscript, I find that the experiments are clearly presented (some questions below) and the performed analysis are generally thorough. Conclusions are rational although, in my opinion, could be further elaborated (see below).

This work has the full potential to be published in nature communications but only after major revision of the language and presentation. It is a pity to submit such work in this format. Most sentences are understandable, with some effort, but the manuscript is difficult to read. Verb tenses are often wrongly used (Literature work should be in the past tense, results should be in the past tense...) and sometimes many tenses are used together in the same sentence. Many typos and words that do not exist in English nor in the chemical vocabulary are scattered in the manuscript (farer ? Franyl?...). This gives the impression that the authors did not care much about their interesting work. I pointed out some mistakes (especially in the SI) but many remain.

Suggestions/Questions:

The authors concluded that ATP-Metal complexes might have served as catalyst in prebiotic reactions and most notably for chiral resolutions. I agree with this very interesting claim that is justified by the results they obtained. The authors however did not extend their view to current biotic reactions. They say for example in the abstract "in biotic reactions ATP-metal complexes function as co-substrates mostly undergoing phosphoanhydride bond cleavage reactions"

Their work showed that ATP-metal complexes could have other functions. This new function i.e. catalysis, could inspire biochemists to look at ATP-metal complexes from another perspective and eventually find that in vivo ATP-metal complexes exist and act as catalysts. I think that the possibility for ATP-metal complexes to have a role in biotic reaction should be mentioned or at least the sentences that describe the current role of ATP should be made less factual. For example : "In biotic reactions ATP-metal complexes **are thought** to function **mostly** as co-substrates undergoing phosphoanhydride bond cleavage reactions."

The catalytic performance of ATP-metal complexes in catalysis should be compared (paragraph or table) with the other artificial metallo-Diels-Alderase (dna based, protein based...) known in literature.

In the abstract : The authors claim to have characterized the catalyst by DFT. DFT is not a characterization technique. Please rephrase.

Introduction, sentence starting with: "Most important for the present work..." the expression "The D-A reaction" should be replaced by "a D-A reaction" otherwise please identify the substrates and the products of the reaction.

Similarly, the sentence starting with : "In a further work Jäschke..." is not clear please identify "the Diels-Alderase", "the bi-molecular reaction" and "both free substrates".

Introduction, the sentence starting with : “Before, the developed ribozyme was able to catalyze “ This sentence is not important and can be removed. Otherwise, it is also not clear and requires explanation especially the identification of the second substrate..

Introduction, sentence starting with: “Systematic experiments based...” again the substrates are not identified, their modification is therefore meaningless...

Introduction, sentence starting with: “Another RNA-based ArMs containing...” what was the catalyzed reaction in this case?

Conclusion : “Since the aldol reaction is also catalyzed by divalent metal ions” needs citation.

All the methods that are described in the manuscript are already described in the SI, why keep both ?

Minor Revisions

“e.e.” should be “*ee*”

“the aza-chalcone” sounds strange, better use “aza-chalcone” without an article.

The word “between” is often used in the wrong context.

Abstract : “chiral scaffolds in metal complexes” should be “chiral scaffolds that bind/surround metal complexes”

Introduction, 1st sentence, “normally” should be replaced by “generally” or “usually”.

Introduction, 1st sentence, “which” should be replaced by “and”.

Introduction, sentence starting with : “In order to...” . The meaning of this sentence is not clear to me, please rephrase it. Probably change the word “clarify” and delete “for enantioselective reactions”.

Introduction, sentence starting with: “In virtue of tunable structures...” The word “formation” should probably be changed to “conformation”?

Introduction, sentence starting with: “Furthermore, a phosphine-modified...” replace “coordinating with a palladium” by “coordinating a palladium”

Scheme 1 : correct “franyl” to “furyl” probably? and add the substitution number for “franyl” and for thienyl.

Several times in the manuscript “xx% e.e. for the *endo* isomer” should be “xx% *ee* in favor of the *endo* isomer”.

Table 1 : *Endo/exo* and *ee* were analyzed from the crude products on a chiral stationary phase. Should be rephrased and “analyzed “ changed to “determined”

Table 1 : All data were conducted at least two individual experiments with the reproducibility of the conversions within $\pm 5\%$, *endo/exo* and e.e. within $\pm 3\%$. Should also be rephrased.

Figure 1 : e.e. should here also be *ee*.

Figure 3 : Rotate both drawing (a) and molecular model (a) to resemble the positioning shown in the catalytic cycle and molecular models (b) and (c)

SI revisions :

Typos:

D-A reaction **between A** and **B**

D-A reaction **of A** and **B**

using **residue** solvent peaks as an internal **standard**

using **residual** solvent peaks as an internal **standards**

The coupling **constant** (J) were reported in hertz (Hz).

The coupling **constants** (J) were reported in hertz (Hz).

Electron paramagnetic resonance (EPR) were conducted

Electron paramagnetic resonance (EPR) **experiments** were conducted

After **stirred** for 30 min at 4 °C,

After **stirring** for 30 min at 4 °C,

The combined organic **solvent**

The combined organic **solvents/layers**

was added **with** the cuvette sealed tightly

was added **and** the cuvette **was** sealed tightly

the line fitted **to decrease** in absorption of 1a versus time,

the line fitted to **the** decrease in absorption of 1a versus time,

are the molar extinction **coefficient** of

are the molar extinction **coefficients** of

UV **titriton** of varied concentrations

UV **titration** of varied concentrations

were determined in comparison **of** the reported

were determined in comparison **with** the reported

The HPLC traces of 3a in racemic form and chiral form **were** shown in Supplementary Figure 21. **In** comparison **of** the reference, we determined that the major product 3a generated by Cu^{2+} -ATP **is** 3a (Si-endo) in (1R,2S,3S,4S) configuration.

The HPLC traces of 3a in racemic form and chiral form are shown in Supplementary Figure 21. By comparison to the reference, we determined that the major product 3a generated by Cu^{2+} -ATP was 3a (Si-endo) in (1R,2S,3S,4S) configuration.

while all TS species have only one imaginary frequency.

while all TS species had only one imaginary frequency.

The corresponding electronic energies (fig 22, fig 23 and fig 24)

The corresponding electronic energies

Difficult to understand (language / chemistry / presentation):

To a solution of ATP (120 mM) in D₂O in a nuclear magnetic tube, Cu^{2+} was titrated to make the final ratio of ATP/ Cu^{2+} over the range from 1000:1 to 5:1.

Language : I am not sure if that is supposed to be :

To a solution of ATP (120 mM) in D₂O in a nuclear magnetic resonance tube, Cu^{2+} was gradually added to vary the ratio of ATP/ Cu^{2+} from 1000:1 to 5:1.

Chemistry: Does Cu^{2+} means copper nitrate? Chloride? triflate? This is the SI the nature of the copper(II) salt must be provided here!

Presentation: Table 2 and table 3 are confusing. Concentrations of ATP, $\text{Cu}(\text{OTf})_2$ and additives are appearing and disappearing in the tables. Sometimes there is a number but sometimes a "-" sometimes "none" and sometime a "space".

This should be clarified. If the concentration is the same in different entries then repeat the number. If the concentration is 0 μM then put 0.

Chemistry: Fig 4, fig 5, fig 7, fig 8, fig 11 Cu^{2+} means copper nitrate, chloride, triflate?

Chemistry: In the typical procedure, paragraph number 4, please mention in which solvent were prepared the "stock solution of ATP" and the "freshly prepared solution of $\text{Cu}(\text{OTf})_2$ ".

Chemistry: In the same paragraph, please state what was the nature of the "short gel column".

Chemistry: paragraph number 5, Cu^{2+} means copper nitrate? Chloride? triflate?

Language : same paragraph, second and third sentences should be combined.

Chemistry : Time of addition of **2** should be stated.

Language: Fig 15, fig 16 and fig 17, the caption should be rephrased. It sounds as if the catalyst (ATP Cu^{2+} , Cu^{2+} , ATP) is carrying out the kinetic plots! The same for fig 18 wherein the buffer is carrying out the kinetic plots.

Chemistry: Fig 16 paragraph number 6, in what solvent ?

Chemistry: paragraph number 8, the reported electronic energies are actually relative to one of the structures (ΔE). This should be clarified and ΔE renamed accordingly "difference in electronic energy" or "relative electronic energy".

Chemistry: paragraph number 8, the method that was used for geometry optimization is described. Please also describe the method that was used for energy calculation.

Reviewer #2 (Remarks to the Author):

The authors have shown for the first time that a Cu(II)-complex of ATP is an active and stereoselective catalyst for the Diels-Alder cycloaddition of an aza-chalcone (substrate 1a) and cyclopentadiene. The latter needs to be used in 200-fold excess relative to 1a. Previous studies have focused on the same reaction, but using completely different biomolecules as scaffolds, as the authors correctly state and cite. The present study includes a huge amount of carefully performed experiments, e.g., use of different Cu(II)-salts, other metals such as Mg, Zn or Ni (which showed drastically less positive results). In the optimized Cu-platform, 90% conversion and about 90% enantioselectivity (ee value) were achieved, and substrate analogs 1b-h resulted in even better catalytic performance. Moreover, ATP analogs were tested for mechanistic/structural purposes, leading to poor results but mechanistic conclusions. Kinetic studies and DFT computations proved to be important in this novel study. It opens the door for the study of other substrates and possibly other reaction types.

Publication in *Nat. Commun.* is recommended following minor revision: The author should present a typical example of the Diels-Alder reaction in which the product is isolated and the respective yield is noted.

Reviewer #3 (Remarks to the Author):

In this manuscript, Wang, Hartig and coworkers describe the application of ATP as chiral ligand for the Cu(II) catalyzed Diels-Alder reaction of cyclopentadiene with a variety of azachalcone derivatives. The reaction proceeds faster in the presence of ATP compared to without and gives rise to moderate to good enantioselectivities of the Diels-Alder product. Interestingly, ATP as ligand proved to be superior to other nucleotides and ADP and AMP, which have one or two phosphate groups. A combined spectroscopic and DFT study was performed to shed light on the interaction, leading to the conclusion that the Cu(II) binds to two of the phosphates and one nitrogen from the nucleobase. Finally, the results are discussed in the context of prebiotic chemistry, proposing a role for ATP in metal catalyzed prebiotic reactions.

The key finding that is presented is that ATP can act as chiral ligand for enantioselective catalysis. In terms of catalysis, this system is of moderate interest since, while the reaction is accelerated by the presence of DNA, it is not faster than other reported DNA/RNA catalyzed versions of this reaction (discussed extensively in the introduction) and the ee's are not spectacular; much higher ee's were obtained using other DNA/RNA-based catalytic systems.

So, it is mainly about the fundamental importance of the ability of ATP to act as ligand for asymmetric catalysis. This is interesting, but for me the most interesting is that while ATP proves to be a good ligand, analogues such as ADP or AMP are not (by the way: the fact that simple nucleotides are not good ligands was already reported before by Boersma et al. *J. Am. Chem. Soc.* 2008, 130, 11783). The role for the phosphate ester groups in ligating the copper ion is surprising.

I am a bit sceptical about the relation to prebiotic chemistry as discussed in the conclusion section. This for me is overinterpretation of the results. Many chiral biomolecules will interact with Cu(II) and give some degree of enantioselectivity in this (and other) reaction. For example, simple amino acids have also been shown to be good ligands for this reaction, giving up to 74% ee (Otto et al. *J. Am. Chem. Soc.* 1999, 121, 29, 679). So it is a bit of stretch to emphasize the role of ATP in these prebiotic asymmetric reactions.

The experimental quality of the work is good and the experimental and supporting information are clear. In table 1 it is stated that all catalysis experiments are the average of 2 independent experiments. I assume this is also the case for the other catalysis results (although this is not mentioned in the captions of other tables). Also the spread in the results is mentioned (calculation of error margins is not possible from 2 experiments).

Point-by-point response

REVIEWER COMMENTS

Reviewer #1 (Remarks to the Author):

The manuscript "A Cu(II)-ATP complex efficiently catalyzes enantioselective Diels-Alder reactions" by Wang et al. describes the preparation and characterization of ATP metal complexes. Catalysis of the Diels-Alder cycloaddition of aza-chlacones and cyclopentadiene by the ATP-Copper(II) complex was efficient and stereoselective. The authors then characterized the ATP-Copper(II) complexation using various techniques and proposed a reaction mechanism that was sustained by DFT calculations.

After carefully examining this manuscript, I find that the experiments are clearly presented (some questions below) and the performed analysis are generally thorough. Conclusions are rational although, in my opinion, could be further elaborated (see below).

This work has the full potential to be published in nature communications but only after major revision of the language and presentation. It is a pity to submit such work in this format. Most sentences are understandable, with some effort, but the manuscript is difficult to read. Verb tenses are often wrongly used (Literature work should be in the past tense, results should be in the past tense...) and sometimes many tenses are used together in the same sentence. Many typos and words that do not exist in English nor in the chemical vocabulary are scattered in the manuscript (farer ? Franyl?...). This gives the impression that the authors did not care much about their interesting work. I pointed out some mistakes (especially in the SI) but many remain.

Response: Thank you very much for the positive evaluation of our work and constructive suggestions. We apologize for the language issues. We have now thoroughly improved the language of the manuscript and supporting information. We have paid additional attention to unclear presentations in the manuscript and supporting information. Moreover, the revised version of the manuscript has been edited by a professional language service and the resulting changes are highlighted in cyan.

Suggestions/Questions:

The authors concluded that ATP-Metal complexes might have served as catalyst in prebiotic reactions and most notably for chiral resolutions. I agree with this very interesting claim that is justified by the results they obtained. The authors however did not extend their view to current biotic reactions. They say for example in the abstract "in biotic reactions ATP-metal complexes function as co-substrates mostly undergoing phosphoanhydride bond cleavage reactions"

Their work showed that ATP-metal complexes could have other functions. This new function i.e. catalysis, could inspire biochemists to look at ATP-metal complexes from another perspective and eventually find that in vivo ATP-metal complexes exist and act as catalysts. I think that the possibility for ATP-metal complexes to have a role in biotic reaction should be mentioned or at least the sentences that describe the current role of ATP should be made less factual. For example: "In biotic reactions ATP-metal complexes **are thought** to function **mostly** as co-substrates undergoing phosphoanhydride bond cleavage reactions."

Response: We have followed your suggestion and revised the sentence in the abstract as suggested "In biotic reactions ATP-metal complexes are thought to function mostly as co-substrates undergoing phosphoanhydride bond cleavage reactions." (page 2, lines 4-5).

The catalytic performance of ATP-metal complexes in catalysis should be compared (paragraph or table) with the other artificial metallo-Diels-Alderase (dna based, protein based...) known in literature.

Response: As suggested, we have added a Supplementary Table 6 to the supporting information to compare the enantioselective catalytic performances of the reported metallo-biohybrid catalysts (see below). In addition, we have added some discussion of the content of the table to the Discussion section (page 13, lines 29-30; page 14, lines 1-2).

Supplementary Table 6 | Comparable catalytic performances of enantioselective Diels-Alder reactions catalyzed by various metallic biohybrid catalysts.

Reference	Publication year	Biological scaffold	Metal species	Conversion (%)	Endo/exo	ee (endo, %)
This work	Not yet	ATP	Cu(OTf) ₂	90	92:8	80
2	1998	L-abrine	Cu(NO ₃) ₂	>90	>90:10	74
3	2000	Ribozyme	NaCl and MgCl ₂	Not mentioned	95:5	95
4	2005	Double-stranded DNA	Cu(NO ₃) ₂ and 9-aminoacridine derivative	>80	91:9	53
5	2006	Double-stranded DNA	Cu(NO ₃) ₂ and 4,4'-dimethyl-2,2'-bipyridine	>80	99:1	99
6	2006	Bovine serum albumin	Copper phthalocyanine-3,4',4'',4'''-tetrasulfonic acid tetrasodium salt	91	91:9	98
7	2009	Bovine pancreatic polypeptide	Cu(NO ₃) ₂	73	>95:5	83
8	2010	tHisF	CuSO ₄	73	93:7	46
9	2010	G-quadruplex DNA	Cu(NO ₃) ₂ and 4,4'-dimethyl-2,2'-bipyridine	>85	95:5	34
10	2012	G-quadruplex DNA	Cu(NO ₃) ₂	99	98:2	74
11	2012	LmrR dimer	Cu(NO ₃) ₂	93	95:5	97
12	2013	SCP-2LV83C	Cu(NO ₃) ₂ and phenanthroline derivative	20	88:22	25
13	2013	G-quadruplex DNA	5,10,15,20-Tetrakis(1-methylpyridinium-4-yl)porphyrinatocopper(II) tetraperchlorate	94	97:3	69
14	2014	Cyclic peptides	Cu(NO ₃) ₂	85	96:4	96
15	2015	Cucurbit[8]uril	Cu(NO ₃) ₂	99	97:3	92
16	2015	Lipase	Cu(NO ₃) ₂ and phenanthroline derivative	98	94:6	92
17	2015	G-quadruplex DNA	Cu(NO ₃) ₂ and terpyridine	>99	98:2	99
18	2016	G-triplex DNA	Cu(NO ₃) ₂	99	99:1	64
19	2017	DNA hairpins	Cu(NO ₃) ₂ and 4,4'-dimethyl-2,2'-bipyridine	99	99:1	96
20	2017	β-Barrel nitrobindin	Cu(NO ₃) ₂	56	95:5	69
21	2017	Double-stranded RNA	Cu(NO ₃) ₂ and 4,4'-dimethyl-2,2'-bipyridine	11	91:9	10
22	2020	c-di-AMP	Cu(OTf) ₂	99	97:3	80

In the abstract: The authors claim to have characterized the catalyst by DFT. DFT is not a characterization technique. Please rephrase.

Response: We have rephrased the description of the DFT calculation as "... proposed the reaction mechanism based on density functional theory (DFT) calculations" (abstract, page 2, lines 9-10).

Introduction, sentence starting with: "Most important for the present work..." the expression "The D-A reaction" should be replaced by "a D-A reaction" otherwise please identify the substrates and the products of the reaction.

Response: As suggested, we have replaced "the D-A reaction" by "a D-A reaction" (page 3, line 25).

Similarly, the sentence starting with: "In a further work Jaschke..." is not clear please identify "the Diels-Alderase", "the bi-molecular reaction" and "both free substrates".

Introduction, the sentence starting with: "Before, the developed ribozyme was able to catalyze" "This sentence is not important and can be removed. Otherwise, it is also not clear and requires explanation especially the identification of the second substrate..."

Introduction, sentence starting with: "Systematic experiments based..." again the substrates are not identified, their modification is therefore meaningless...

Response: As suggested, we have pointed out the substrates and rephrased the mentioned sentences as follows: "In a further work Jaeschke and co-workers developed a 49-mer Diels-Alderase ribozyme that could catalyse the D-A reaction of anthracene derivatives and maleimides with an enantiomeric excess (ee) of up to 95%.⁴⁰ Systematic experiments based on the anthracene derivative dienes and of maleimide dienophiles suggest...." (page 3, lines 27-30).

Introduction, sentence starting with: "Another RNA-based ArMs containing..." what was the catalyzed reaction in this case?

Response: We have added the reaction type in the sentence "Another RNA-based ArMs containing either dsRNA or a hairpin RNA exhibited very low enantioselective induction in a D-A reaction compared with the corresponding DNA-based ArMs." (page 4, lines 9-11).

Conclusion: "Since the aldol reaction is also catalyzed by divalent metal ions" needs citation.

Response: We have added several references to support our discussion (refs 62-65, page 14, line 18; page 19, lines 10-17).

All the methods that are described in the manuscript are already described in the SI, why keep both?

Response: We have updated the method section in the manuscript (Methods, pages 14-15) and deleted the redundant methods description in the supporting information.

Minor Revisions

"e.e." should be "ee"

Response: We have changed "e.e." to "ee" through the whole manuscript and supporting information.

“the aza-chalcone” sounds strange, better use “aza-chalcone” without an article.

Response: As suggested, we have deleted the unnecessary article before “azachalcone” in the revised manuscript (page 7, lines 9, 12, 20).

The word “between” is often used in the wrong context.

Response: We have revised the inappropriate use of “between” in the manuscript as follows:

(1) Page 10, line 8: “To assess the binding affinity between ATP and Cu²⁺ ions...” was revised to “To assess the binding affinity of the Cu(II)-ATP complex...”.

(2) Page 10, lines 9-10: “The k_b between ATP and Cu²⁺ ions...” was revised to “The k_b value for ATP and Cu²⁺ ions...”.

(3) Page 10, line 12, the caption of Fig. 2: “The apparent binding constant (k_b) between ATP and Cu²⁺ ions...” was revised to “The apparent binding constant (k_b) of ATP and Cu²⁺ ions...”.

(4) Page 13, lines 28-29: “an intramolecular hydrogen bond occurs between the 6-amino and γ -phosphate oxygen moieties...” was revised to “an intramolecular hydrogen bond between the 6-amino and γ -phosphate oxygen moieties...”.

Abstract: “chiral scaffolds in metal complexes” should be “chiral scaffolds that bind/surround metal complexes”

Response: We agreed with your comment and revised “chiral scaffold in metal complexes” to “chiral scaffolds that bind/surround metal complexes” in the abstract. (page 2, lines 1-2).

Introduction, 1st sentence, “normally” should be replaced by “generally” or “usually”.

Introduction, 1st sentence, “which” should be replaced by “and”.

Response: As suggested, we have revised the 1st sentence in the introduction as “Artificial metalloenzymes (ArMs) are usually constructed by embedding metal cofactors into the chiral scaffolds of biological molecules that are used to expand the reaction types and unearth novel functions of the biomolecules.” (page 3, lines 1-3).

Introduction, sentence starting with: “In order to...”. The meaning of this sentence is not clear to me, please rephrase it. Probably change the word “clarify” and delete “for enantioselective reactions”.

Response: We have rephrased the sentence as “To precisely characterize the active centres and obtain insight into the reaction mechanisms of ArMs, simple scaffolds of peptides and amino acids have been employed to rationally design artificial metallo-peptides and metallo-amino acids.” (page 3, lines 6-9).

Introduction, sentence starting with: “In virtue of tunable structures...” The word “formation” should probably be changed to “conformation”?

Response: We have followed your suggestion and changed “formation” to “conformation” in the revised sentence (page 3, line 19).

Introduction, sentence starting with: “Furthermore, a phosphine-modified...” replace “coordinating with a palladium” by “coordinating a palladium”

Response: As suggested, we have replaced “coordinating with a palladium” by “coordinating a

palladium" (page 4, lines 16-17).

Scheme 1: correct "franyl" to "furyl" probably? and add the substitution number for "franly" and for thienyl.

Response: Sorry for this mistake. We have corrected "franyl" to "furyl" and added the substitution numbers for furyl and thienyl in scheme 1 (page 5).

Several times in the manuscript "xx% e.e. for the *endo* isomer" should be "xx% ee in favor of the *endo* isomer".

Response: We have followed your suggestion and revised "xx% e.e. for the *endo* isomer" to "xx% ee in favour of the *endo* isomer". In addition, the similar description of the "*exo* isomer" was also revised. The detailed revisions are listed as follows:

(1) Page 5, lines 22-23: "...and 70% ee in favour of the *endo* isomer..."

(2) Page 7, lines 15-16: "...93% ee in favour of the *exo* isomer of 3e..."

(3) Page 7, line 17: "...an 84% ee in favour of the *endo* isomer of 3c..."

(4) Page 7, line 24: "...65% in favour of the *endo* isomer..."

Table 1: *Endo/exo* and *ee* were analyzed from the crude products on a chiral stationary phase. Should be rephrased and "analyzed" changed to "determined"

Response: We have changed "analyzed" to "determined" and rephrased the corresponding sentence to "*Endo/exo* and *ee* were determined from the crude products by chiral HPLC" (page 6, lines 4-5, the footnote of Table 1).

Table 1: All data were conducted at least two individual experiments with the reproducibility of the conversions within $\pm 5\%$, *endo/exo* and e.e. within $\pm 3\%$. Should also be rephrased.

Response: We have rephrased the sentence "All data were the averages of at least two individual experiments (reproducibility: $\pm 5\%$ conversion, $\pm 3\%$ *endo/exo* and $\pm 3\%$ *ee*)" (page 6, lines 5-6, the footnote of Table 1).

Figure 1: e.e. should here also be *ee*.

Response: We have changed "e.e." to "*ee*" in Figure 1 (page 9).

Figure 3: Rotate both drawing (a) and molecular model (a) to resemble the positioning shown in the catalytic cycle and molecular models (b) and (c)

Response: As suggested, we have rotated both drawing and molecular model in Fig. 3a to resemble the positioning shown in Fig. 3b-c (page 12, Figure 3).

SI revisions:

Typos:

D-A reaction between A and B

D-A reaction of A and B

Response: As suggested, we have revised the inappropriate use of "between" in the captions of Supplementary Figs. 15-18 (pages S14-S15 in the SI).

using **residue** solvent peaks as an internal **standard**
using **residual** solvent peaks as an internal **standards**

Response: As suggested, we have revised “using residue solvent peaks as an internal standard” to “using residual solvent peaks as internal standards” (page S3 in the SI, lines 7-8).

The coupling **constant** (J) were reported in hertz (Hz).
The coupling **constants** (J) were reported in hertz (Hz).

Response: As suggested, we have revised “The coupling constant (J) were reported in hertz (Hz).” to “The coupling constants (J) were reported in hertz (Hz).” (page S3 in the SI, lines 10-11).

Electron paramagnetic resonance (EPR) were conducted
Electron paramagnetic resonance (EPR) **experiments** were conducted

Response: As suggested, we have revised “Electron paramagnetic resonance (EPR) were conducted...” to “Electron paramagnetic resonance (EPR) experiments were conducted...” (page S3 in the SI, line 14).

After **stirred** for 30 min at 4 °C,
After **stirring** for 30 min at 4 °C,
The combined organic **solvent**
The combined organic **solvents/layers**

Response: Since the method of “typical procedure” was moved from the supporting information to the Methods section in the manuscript, we have revised the above sentences as follows:

- (1) Page 14, lines 27-28: “After stirring for 30 min at 4 °C”**
- (2) Page 15, line 1: “The combined organic layers”**

was added **with** the cuvette sealed tightly
was added **and** the cuvette **was** sealed tightly
the line fitted to **decrease** in absorption of 1a versus time,
the line fitted to **the** decrease in absorption of 1a versus time,
are the molar extinction **coefficient** of
are the molar extinction **coefficients** of

Response: Since the method of “kinetic measurements” was moved from the supporting information to the Methods section in the manuscript, we have revised the above sentences as follows:

- (1) Page 15, lines 14-15: “Followed by an immediate addition of 2 (final conc. 5 mM), the measurement was started and the cuvette was sealed tightly.”**
- (2) Page 15, line 16: “the line fitted to the decrease in absorption of 1a versus time”**
- (3) Page 15, lines 20-21: “are the molar extinction coefficients of”**

UV **titriton** of varied concentrations
UV **titration** of varied concentrations

Response: As suggested, we have revised “UV titriton of varied concentrations” to “UV titration of varied concentrations” (page S16 in the SI, the caption of Supplementary Fig. 19).

were determined in comparison of the reported
were determined in comparison with the reported

Response: As suggested, we have revised “were determined in comparison of the reported” to “were determined in comparison with the reported” (page S17 in the SI, lines 4-5).

The HPLC traces of 3a in racemic form and chiral form were shown in Supplementary Figure 21. In comparison of the reference, we determined that the major product 3a generated by Cu^{2+} -ATP is 3a (Si-endo) in (1R,2S,3S,4S) configuration.

The HPLC traces of 3a in racemic form and chiral form are shown in Supplementary Figure 21. By comparison to the reference, we determined that the major product 3a generated by Cu^{2+} -ATP was 3a (Si-endo) in (1R,2S,3S,4S) configuration.

Response: As suggested, we have revised “The HPLC traces of 3a in racemic form and chiral form were shown in Supplementary Figure 21. In comparison of the reference, we determined that the major product 3a generated by Cu^{2+} -ATP is 3a (Si-endo) in (1R,2S,3S,4S) configuration” to “The HPLC traces of 3a in racemic form and chiral form are shown in Supplementary Fig. 21. By comparison to the reference, we determined that the major product 3a generated by Cu^{2+} -ATP was 3a (Si-endo) in (1R,2S,3S,4S) configuration” (page S16 in the SI, lines 9-11).

while all TS species have only one imaginary frequency.

while all TS species had only one imaginary frequency.

Response: Since the method of “DFT calculation” was moved from the supporting information to the Methods section in the manuscript, we have revised “while all TS species have only one imaginary frequency” to “whereas all TS species had only one imaginary frequency” (page 15, line 35).

The corresponding electroic energies (fig 22, fig 23 and fig 24)

The corresponding electronic energies

Response: We have revised all the “electronic energies” to “electronic energies” (pages S18-S19 in the SI, the captions of Supplementary Figs. 22-25).

Difficult to understand (language / chemistry / presentation):

To a solution of ATP (120 mM) in D₂O in a nuclear magnetic tube, Cu^{2+} was titrated to make the final ratio of ATP/ Cu^{2+} over the range from 1000:1 to 5:1.

Language: I am not sure if that is supposed to be:

To a solution of ATP (120 mM) in D₂O in a nuclear magnetic resonance tube, Cu^{2+} was gradually added to vary the ratio of ATP/ Cu^{2+} from 1000:1 to 5:1.

Chemistry: Does Cu^{2+} means copper nitrate? Chloride? triflate? This is the SI the nature of the copper(II) salt must be provided here!

Response: We apologize for this unclear presentation and thank you very much for your suggestion. We have revised the unclear sentence to “To a solution of ATP (120 mM) in D₂O in a nuclear magnetic tube, a stock solution of CuCl_2 in D₂O was gradually added to vary the ratio of ATP/ Cu^{2+} from 1000:1 to 5:1.” (page S2 in the SI, lines 12-13).

To show the nature of copper(II) salts in the SI, we have revised “ Cu^{2+} ions” to “ $\text{Cu}(\text{OTf})_2$ ” (pages S8-S12, S14 in the SI, the captions of Supplementary Figs. 4-5, 7-12, 15-16).

Presentation: Table 2 and table 3 are confusing. Concentrations of ATP, Cu(OTf)₂ and additives are appearing and disappearing in the tables. Sometimes there is a number but sometimes a “-” sometimes “none” and sometime a “space”.

This should be clarified. If the concentration is the same in different entries then repeat the number. If the concentration is 0 μM then put 0.

Response: Sorry for the unclear symbols, we have revised Supplementary Tables 2 and 3 with clarified meanings (pages S3-S4).

Chemistry: Fig 4, fig 5, fig 7, fig 8, fig 11 Cu²⁺ means copper nitrate, chloride, triflate?

Response: The “Cu²⁺ ions” in the Supplementary Figs. 4, 5, 7, 8 and 11 are “Cu(OTf)₂”. We have revised the unspecific “Cu²⁺ ions” to “Cu(OTf)₂” (pages S8-S12 in the SI, the captions of Supplementary Figs. 4-5, 7-12).

Chemistry: In the typical procedure, paragraph number 4, please mention in which solvent were prepared the “stock solution of ATP” and the “freshly prepared solution of Cu(OTf)₂”.

Chemistry: In the same paragraph, please state what was the nature of the “short gel column”.

Response: In the “typical procedure” in the revised manuscript, we have clarified the solvents for the solutions of ATP and Cu(OTf)₂ as “a stock solution of ATP in water (final conc. 250 μM) and a freshly prepared aqueous solution of Cu(OTf)₂...” (page 14, lines 25-26).

To describe the details of the “short gel column”, we have added a short note following the term of short gel column stating “a 5 cm length of glass dropper was filled with the silica gel to a height of ca. 2 cm with some cotton at the bottom” (page 14, line 31; page 15, line 1).

Chemistry: paragraph number 5, Cu²⁺ means copper nitrate? Chloride? triflate?

Language: same paragraph, second and third sentences should be combined.

Chemistry: Time of addition of **2** should be stated.

Response: In paragraph number 5 in the SI, the Cu²⁺ ions originate from Cu(OTf)₂.

Since the paragraph number 5 of “kinetic assay” was moved from the supporting information to the Methods section in the manuscript, we have combined the second and third sentence in the revised manuscript as “ATP (final conc. 250 μM) in an MES buffer (20 mM, pH 5.5) was added to a 2 mL quartz cuvette containing a small magnet and stirred for 10 min, and then an aqueous solution of Cu(OTf)₂ (final conc. 50 μM) was added.” (page 15, lines 11-13).

We have pointed out the addition time of **2 in the revised manuscript as “Followed by an immediate addition of **2** (final conc. 5 mM), the measurement was started and the cuvette was sealed tightly.” (page 15, lines 14-15).**

Language: Fig 15, fig 16 and fig 17, the caption should be rephrased. It sounds as if the catalyst (ATP Cu²⁺, Cu²⁺, ATP) is carrying out the kinetic plots! The same for fig 18 wherein the buffer is carrying out the kinetic plots.

Response: Sorry for this misunderstanding. We have rephrased the captions of Supplementary Figs. 15-18 as follows (page S14-S15 in the SI):

Supplementary Fig. 15 | Kinetic plots of Cu²⁺-ATP catalyzed D-A reactions by monitoring the disappearance of the UV absorbance of 1a at 326 nm. Reaction conditions: 1a (20, 30, 50 μM), **2 (5 mM), ATP (250 μM), Cu(OTf)₂ (50 μM), MES buffer (2000 μL, 20 mM, pH 5.5), 4 °C.**

Supplementary Fig. 16 | Kinetic plots of Cu(OTf)₂ catalyzed D-A reactions by monitoring the disappearance of the UV absorbance of 1a at 326 nm. Reaction conditions: 1a (20, 30, 50 μM), 2 (5 mM), Cu(OTf)₂ (50 μM), MES buffer (2000 μL, 20 mM, pH 5.5), 4 °C.

Supplementary Fig. 17 | Kinetic plots of ATP catalyzed D-A reactions by monitoring the disappearance of the UV absorbance of 1a at 326 nm. Reaction conditions: 1a (20, 30, 50 μM), 2 (5 mM), ATP (250 μM), MES buffer (2000 μL, 20 mM, pH 5.5), 4 °C.

Supplementary Fig. 18 | Kinetic plots of the D-A reactions in the absence of catalysts by monitoring the disappearance of the UV absorbance of 1a at 326 nm. Reaction conditions: 1a (20, 30, 50 μM), 2 (5 mM), MES buffer (2000 μL, 20 mM, pH 5.5), 4 °C.

Chemistry: Fig 16 paragraph number 6, in what solvent?

Response: In the paragraph number 6 (Note: the number 6 is changed to number 4 in the revised SI), the solvent is H₂O and we have added the solvent information in the corresponding descriptions. (page S16 in the SI, lines 3 and 11; the caption of Supplementary Fig. 19)

Chemistry: paragraph number 8, the reported electronic energies are actually relative to one of the structures (ΔE). This should be clarified and ΔE renamed accordingly “difference in electronic energy” or “relative electronic energy”.

Response: As suggested, we have added the illustration of the reported energies and their relative standards as “In order to compare the stability among all candidates, the relative electronic energies (ΔE) were listed in Supplementary Figs. 22-25 in which the most stable configuration was set as the zero point.” (page S18 in the SI, lines 2-3).

Chemistry: paragraph number 8, the method that was used for geometry optimization is described. Please also describe the method that was used for energy calculation.

Response: Since the description of the DFT calculation was moved from the supporting information to the Methods section in the manuscript, we have added the method for the calculation of ΔE as “The kinetic barriers of the non-catalyst reaction and the catalytic reaction with the most stable precursor were evaluated by calculating the single-point energies at the M06-2X-D3/LANL2DZ~6-311G(d,p) level.” (page 15, lines 32-34).

Reviewer #2 (Remarks to the Author):

The authors have shown for the first time that a Cu(II)-complex of ATP is an active and stereoselective catalyst for the Diels-Alder cycloaddition of an azachalcone (substrate 1a) and cyclopentadiene. The latter needs to be used in 200-fold excess relative to 1a. Previous studies have focused on the same reaction, but using completely different biomolecules as scaffolds, as the authors correctly state and cite. The present study includes a huge amount of carefully performed experiments, e.g., use of different Cu(II)-salts, the metals such as Mg, Zn or Ni (which showed drastically less positive results). In the optimized Cu-platform, 90% conversion and about 90% enantioselectivity (ee value) were achieved, and substrate analogs 1b-h resulted in even better catalytic performance. Moreover, ATP analogs were tested for mechanistic/structural purposes, leading to poor results but mechanistic conclusions. Kinetic studies and DFT computations proved to be important in this novel study. It opens the door for the study of other

substrates and possibly other reaction types.

Publication in Nat. Commun. is recommended following minor revision: The author should present a typical example of the Diels-Alder reaction in which the product is isolated and the respective yield is noted.

Response: Thank you very much for the high evaluation of our work. Since the Cu(II)-ATP catalyzed Diels-Alder reactions were conducted in analytical scales in the manuscript, we obtained an isolated yield of 75% by combining 20 individual Diels-Alder reactions. In addition, the Diels-Alder reaction was carried out in a preparative scale using 1a of 105 mg. We obtained the product 3a with an isolated yield of 80% and an ee value of 65%. (page 7, lines 21-24).

Reviewer #3 (Remarks to the Author):

In this manuscript, Wang, Hartig and coworkers describe the application of ATP as chiral ligand for the Cu(II) catalyzed Diels-Alder reaction of cyclopentadiene with a variety of azachalcone derivatives. The reaction proceeds faster in the presence of ATP compared to without and gives rise to moderate to good enantioselectivities of the Diels-Alder product. Interestingly, ATP as ligand proved to be superior to other nucleotides and ADP and AMP, which have one or two phosphate groups. A combined spectroscopic and DFT study was performed to shed light on the interaction, leading to the conclusion that the Cu(II) binds to two of the phosphates and one nitrogen from the nucleobase. Finally, the results are discussed in the context of prebiotic chemistry, proposing a role for ATP in metal catalyzed prebiotic reactions.

The key finding that is presented is that ATP can act as chiral ligand for enantioselective catalysis. In terms of catalysis, this system is of moderate interest since, while the reaction is accelerated by the presence of DNA, it is not faster than other reported DNA/RNA catalyzed versions of this reaction (discussed extensively in the introduction) and the ee's are not spectacular; much higher ee's were obtained using other DNA/RNA-based catalytic systems.

So, it is mainly about the fundamental importance of the ability of ATP to act as ligand for asymmetric catalysis. This is interesting, but for me the most interesting is that while ATP proves to be a good ligand, analogues such as ADP or AMP are not (by the way: the fact that simple nucleotides are not good ligands was already reported before by Boersma et al. J. Am. Chem. Soc. 2008, 130, 11783). The role for the phosphate ester groups in ligating the copper ion is surprising.

Response: We agree with the reviewer's comment that the fundamental importance of our work is ATP serving as an efficient ligand for enantioselective catalysis rather than ADP or AMP. We have added further discussion of the efficient but simple ligand of ATP by comparison to reported metallo-biohybrid catalysts (page 13, lines 29-20; page 14, lines 1-6).

I am a bit sceptical about the relation to prebiotic chemistry as discussed in the conclusion section. This for me is overinterpretation of the results. Many chiral biomolecules will interact with Cu(II) and give some degree of enantioselectivity in this (and other) reaction. For example, simple amino acids have also been shown to be good ligands for this reaction, giving up to 74% ee (Otto et al. J. Am. Chem. Soc. 1999, 121, 29, 679). So it is a bit of stretch to emphasize the role of ATP in these prebiotic asymmetric reactions.

Response: We agree with the reviewer that many chiral biomolecules will interact with Cu(II) ions and give some degree of enantioselectivity in this Diels-Alder reaction. For additional clarity

we have added the mentioned Supplementary Table 6 that compares the metallic biohybrid catalysts reported in the literature (page 13, lines 29-30; page 14, lines 1-6). In order to address the raised concern that the presented Cu²⁺-ATP-catalyzed D-A reactions utilizing an azachalcone and cyclopentadiene are likely not of high importance in prebiotic asymmetric reactions, we have rephrased the discussion of the implications for prebiotic chemistry in a way that we only imply that there could be a possible role of ATP complexes in prebiotically relevant reactions such as the Aldol reaction (page 14, line 7-22).

The experimental quality of the work is good and the experimental and supporting information are clear. In table 1 it is stated that all catalysis experiments are the average of 2 independent experiments. I assume this is also the case for the other catalysis results (although this is not mentioned in the captions of other tables). Also the spread in the results is mentioned (calculation of error margins is not possible from 2 experiments).

Response: Thank you very much for the positive evaluation of our work. In Table 1, we stated that all reactions were conducted at least two times in individual experiments (in fact some experiments were conducted three times) to demonstrate the reproducibility of our catalytic results. In order to clarify the reproducibility of other catalysis results, we have added an explanation to the caption of Fig. 1 (page 9, lines 4-5).

We agree that the calculation of error bars is not possible from two experiments. For the kinetic studies for calculating k_{app} we have carried out three individual measurements and standard deviations are shown in Table 2. We have added a corresponding description to the caption of Table 2 (page 8, lines 3-4).

REVIEWERS' COMMENTS:

Reviewer #1 (Remarks to the Author):

The authors meticulously corrected the manuscript in compliance with the reviewer's comments.

In my opinion the manuscript can be published after repairing table 6 in the SI as well as the associated references.

This table compares the artificial enzyme of this manuscript with other artificial metallo-Diels Alderases that uses 2-azachalcone and cyclopentadiene as substrates. The table is however far from being comprehensive. Many examples are missing, for instance some of the work of Hayashi et al. (on fhuA), some of the work of Roelfes et al. (on bpp), the work of Eppinger et al. (on mTFP), and surprisingly all the work of Mahy et al..

In the references, sometimes all authors names are displayed while other times a selective or arbitrary? number of author names are displayed...

Point-by-point response

REVIEWERS' COMMENTS:

Reviewer #1 (Remarks to the Author):

The authors meticulously corrected the manuscript in compliance with the reviewer's comments. In my opinion the manuscript can be published after repairing table 6 in the SI as well as the associated references.

This table compares the artificial enzyme of this manuscript with other artificial metallo-Diels Alderases that uses 2-azachalcone and cyclopentadiene as substrates. The table is however far from being comprehensive. Many examples are missing, for instance some of the work of Hayashi et al. (on fhuA), some of the work of Roelfes et al. (on bpp), the work of Eppinger et al. (on mTFP), and surprisingly all the work of Mahy et al..

In the references, sometimes all authors names are displayed while other times a selective or arbitrary? number of author names are displayed...

Response: Thank you very much for your comments and suggestions. We have added more works of artificial metallo-Diels-Alderase using 2-azachalcone and cyclopentadiene as substrates in Supplementary Table 6 (Biological scaffold: ref 17: β -Barrel protein nitrobindin, ref 18: FhuA, ref 19: Neocarzinostatin, ref 23: α Rep A3, ref 25: HEK-A24, ref 26: mTFP, ref 27: α Rep (A3_A3')F119C, ref 28: ACCO).

The discrepancy of the authors names that displaying in the references is due to the standard Nature referencing style, which requires all authors are shown in reference lists unless there are six or more, otherwise only the first author is given followed by 'et al.'.